# Photonics of Trimethine Cyanine Dyes as Probes for Biomolecules

**DOI:** 10.3390/molecules27196367

**Published:** 2022-09-27

**Authors:** Pavel G. Pronkin, Alexander S. Tatikolov

**Affiliations:** N.M. Emanuel Institute of Biochemical Physics, Russian Academy of Sciences, 4 Kosygin Str., 119334 Moscow, Russia

**Keywords:** trimethine cyanine dyes, biomolecules, DNA, albumins, photonics, spectral-fluorescent properties, isomerization, triplet state

## Abstract

Cyanine dyes are widely used as fluorescent probes in biophysics and medical biochemistry due to their unique photophysical and photochemical properties (their photonics). This review is focused on a subclass of the most widespread and studied cyanine dyes—trimethine cyanines, which can serve as potential probes for biomolecules. The works devoted to the study of the noncovalent interaction of trimethine cyanine dyes with biomolecules and changing the properties of these dyes upon the interaction are reviewed. In addition to the spectral-fluorescent properties, elementary photochemical properties of trimethine cyanines are considered, including: photoisomerization and back isomerization of the photoisomer, generation and decay of the triplet state, and its quenching by oxygen and other quenchers. The influence of DNA and other nucleic acids, proteins, and other biomolecules on these properties is covered. The interaction of a monomer dye molecule with a biomolecule usually leads to a fluorescence growth, damping of photoisomerization (if any), and an increase in intersystem crossing to the triplet state. Sometimes aggregation of dye molecules on biomolecules is observed. Quenching of the dye triplet state in a complex with biomolecules by molecular oxygen usually occurs with a rate constant much lower than the diffusion limit with allowance for the spin-statistical factor 1/9. The practical application of trimethine cyanines in biophysics and (medical) biochemistry is also considered. In conclusion, the prospects for further studies on the cyanine dye–biomolecule system and the development of new effective dye probes (including probes of a new type) for biomolecules are discussed.

## 1. Introduction

Cyanine (polymethine) dyes are currently widely used in various fields of science and technology [1,2,3]. This is due to the peculiarities of their electronic structure and molecular arrangement, which determine the unique photophysical and photochemical properties of the dyes. The cyanine chromophore is a polymethine chain with conjugated –C=C– bonds, which usually link terminal heterocycles (thiazoles, oxazoles, etc.), and an odd number of C atoms, the length of which mainly determines the spectral region of absorption and emission of the dyes. The chromophore moiety of cyanines usually carries a total positive charge. In symmetric cyanines, the electron density is uniformly distributed over the C–C bonds of the polymethine chain (with the order of bonds approaching one and a half), whereas the charges on the C atoms alternate [4]. Depending on the length of the polymethine chain, cyanine dyes are divided into monomethine cyanines (with a polymethine “chain” consisting of only one methylene group, –CH=), trimethine cyanines (or carbocyanines, with a polymethine chain consisting of three methylene groups, –CH=CH–CH=), pentamethine cyanines (or dicarbocyanines, with a five-methylene polymethine chain, –CH=CH–CH=CH–CH=), heptamethine cyanines (or tricarbocyanines, with a seven-methylene polymethine chain, –CH=CH–CH=CH–CH=CH–CH=), etc. Elongation of the polymethine chain by one –CH=CH– unit causes a long-wavelength shift of the absorption spectra of cyanines (the so-called vinylene shift). For symmetric cyanines, this shift is usually 100–130 nm [4]. The whole variety of cyanine dyes covers a wide spectral absorption range from near UV to near IR, providing high extinction coefficients.

The synthesis of cyanines of various structures is well-developed, which makes it possible to obtain the dyes of the desired structure (see, e.g., [5,6,7,8,9]). For the synthesis, quaternary ammonium salts with active methyl groups at 2 and/or 4 positions, such as salts of methylpyridines and methylquinolines, are often used. To increase the reactivity of the methyl group, other heterocyclic quaternary salts can be used, such as salts of 2-methyl thiazole, 2-methyl benzothiazole, 2-methyl oxazole, 2-methyl benzoxazole, 2-methyl selenazole, and 2-methyl benzoselenazole. Additionally, quaternary salts with an active hydrogen atom at 4 and/or 1 positions are used, such as salts of pyridine, quinoline, and isoquinoline [2].

The flexibility of the polymethine chain provides the possibility of vibrations, torsions, and rotations of its fragments, which in turn creates channels for the nonradiative dissipation of the energy of the electronically excited state (*S*_1_) of the cyanine molecule. This creates a dependence of the photophysical properties of cyanines (the lifetime of the *S*_1_ state, the fluorescence quantum yield) on the molecular environment (in solutions, primarily on the solvent viscosity). This also creates the possibility of isomerization of the molecule by rotation around one of the C–C bonds of the polymethine chain (*trans*-to-*cis* or *cis*-to-*trans* isomerization), which is also one of the channels of nonradiative deactivation of the electronic excitation energy of the dye.

Taking into account that molecules of cyanine dyes without substituents in the polymethine chain are in the all-*trans* form (denoted below as *S*_0_(*trans*)) [10], the primary photophysical and photochemical processes that can occur upon photoexcitation of cyanine molecules (i.e., their photonics) may be represented by the following scheme of competitive pathways (Figure 1):

The photoisomer formed (*S*_0_(*cis*)) is usually unstable and undergoes dark back isomerization, reverting to the initial *trans*-isomer (*S*_0_(*trans*)). Note that the quantum yield of intersystem crossing to the triplet state (*S*_1_ → *T*) for trimethine cyanine dyes in solution is usually very low (<0.05) [11].

Since the photonics of cyanine dyes depends on the molecular environment, they could serve as potential probes in the study of various molecularly organized media. A prerequisite for the use of cyanines as probes for biomolecules is the data on their noncovalent binding to various biomacromolecules, which is accompanied by an increase in fluorescence intensity [12]. The fluorescence growth is a consequence of an increase in the rigidity of the dye molecule in the complex with a biomolecule, which leads to suppression of the nonradiative processes of internal conversion and photoisomerization.

Another important feature of cyanine dyes is the ability to form aggregates (self-associates of monomer dye molecules)—dimers and H- and J-aggregates—which is most pronounced in aqueous solutions and has a great impact on cyanine photonics. In particular, the absorption bands of H-aggregates and dimers are shifted hypsochromically with respect to the monomer band, and they usually do not fluoresce or fluoresce weakly. At the same time, J-aggregates have narrow absorption bands lying in the longer-wavelength region and exhibit fluorescence with a very small Stokes shift [13,14]. The formation of disordered aggregates is also observed [15]. The aggregation can be stimulated by biomolecules [16].

Our review is focused on a subclass of the most widespread and studied cyanine dyes—trimethine cyanines, whose generalized structure is as follows (Figure 1):

Where R, R_1_, and R_2_ are substituents (R is a *meso*- (or 9-) substituent; for *meso*-unsubstituted dyes R = H), and X is a heteroatom.

The generalized scheme of *trans*–*cis* photoisomerization and thermal back isomerization of the photoisomer of trimethine cyanine dyes is shown in Figure 2.

Cyanine dyes are characterized by an equilibrium between the *trans*- and *cis*-isomers [17,18]. In the case of cyanine dyes that do not have substituents in the polymethine chain, the *trans*-isomers are more energetically stable, and in solutions, the dyes are in this form [10]. For thiacarbocyanine dyes having *meso*-substituents in the polymethine chain, a dynamic (depending on the solvent polarity) equilibrium between the *cis*- and *trans*-isomers is observed. Under the steric influence of *meso*-substituents, the structure of the *trans*-isomers of the dyes is distorted (becomes nonplanar), which leads to an increase in their energy and approaches the energy of *cis*-isomers [18]. In nonpolar and low-polarity solvents, the equilibrium is shifted toward the *trans*-isomer, whereas in polar solvents, this shift occurs toward the *cis*-isomer, which has very weak fluorescence [18,19].

The main data on spectral-fluorescent and photochemical properties of trimethine cyanines in solutions are summarized in [4,20,21]. In this review, we consider the influence of noncovalent interaction with biomolecules on the spectral-fluorescent properties and primary photochemical processes (isomerization, generation, and quenching of the triplet state) in molecules of trimethine cyanine dyes—potential probes for biomolecules, as well as on their aggregation properties.

This review attempts to show the diversity of structures of trimethine cyanine dyes that could be used as probes for biomolecules and, where possible, the relationship between these structures and properties important for biomolecular probes.

## 2. Influence of Interaction with DNA and Other Nucleic Acids on the Spectral-Fluorescent Properties of Trimethine Cyanine Dyes

The noncovalent interaction of cationic trimethine cyanine dyes of various structures with dsDNA molecules has been studied in many works (see, e.g., [22,23]).

In the noncovalent interaction of dyes with DNA, hydrophobic interactions, hydrogen bonds, and Coulomb attractive forces play a stabilizing role. Due to the presence of negatively charged residues of phosphate groups, molecules of DNA and other nucleic acids in solutions are polyanions. Ligand dyes that effectively interact with DNA should have positive charges. Classical cationic trimethine cyanine dyes meet this requirement.

Upon the interaction of monomeric ligands with the DNA double helix, two main types of binding are possible: the formation of intercalation complexes (the ligand intercalates between base pairs of the DNA molecule) and “external” binding (for example, binding in the minor groove of the DNA helix) [23] (Figure 3). Intercalation is typically observed for cationic molecules with planar aromatic rings. This binding mode requires two adjacent base pairs to separate from one another to create a binding pocket for the ligand. Minor groove binders, on the other hand, usually should have some flexibility since this allows the molecule to adjust its structure to follow the groove as it twists around the central axis of the helix. Binding in the minor groove requires substantially less distortion of the DNA compared with the intercalative binding [23]. In general, the intercalative complexes of cyanine dyes are characterized by relatively low equilibrium constants; these complexes are insensitive to an increase in the ionic strength. Complexes in the minor groove of the biopolymer double helix can consist of both monomeric cyanine molecules and dye aggregates. This type of complex is characterized by higher equilibrium constants and sensitivity to ionic strength [24].

The most characteristic effect observed upon the interaction of a dye molecule with nucleic acids is a bathochromic shift in the absorption spectra. This shift is caused by changes in the solvate shell of dye molecules during complex formation, in which new components are included in the hydrate shell of the dye molecule, namely, parts of some structural units of DNA, which can be comparable to the effect of solvent replacement. Hence, one of the stages of the DNA binding process is the transfer of dye molecules from the polar phase of the solvent (water) to the less polar phase of DNA molecules, which is energetically favorable. Changes in the absorption and fluorescence spectra of a trimethine cyanine dye in the presence of DNA and the fluorescence growth are shown in Figure 4a,b.

Along with the interaction in the monomeric form, cyanine dyes can assemble into ordered supramolecular aggregates (in particular, H- and J-aggregates) on a DNA macromolecule as a template [23], which can be used for structural studies of nucleic acid molecules. In many cases, cyanine dyes are also characterized by the concomitant formation of disordered aggregates both by themselves and on DNA and other biomacromolecules, which usually hinders the use of the dyes as biomolecular probes.

As mentioned above, thiacarbocyanine dyes with meso- (or 9-) substituents in the polymethine chain are characterized by a dynamic (depending on the molecular environment) equilibrium between the cis- and trans-isomers. This equilibrium is also affected by complexation with biomolecules. Since the photophysical and photochemical properties of these isomers are very different (in particular, the fluorescence of the cis-isomers, in the form of which these dyes are present in aqueous media, is very weak, whereas the trans-isomers fluoresce), this creates additional advantages for using meso-substituted thiacarbocyanines as fluorescent probes for biomolecules [25].

Cationic trimethine cyanine dyes **1**–**13** (having substituents in the polymethine chain and in terminal heterocycles; Figure 5) have been proposed as spectral-fluorescent probes for nucleic acids [24,26,27]. In the presence of nucleic acids, an increase in dye fluorescence is observed. Suggested for use are thiacarbocyanines **4**–**13**, which contain various substituents in the 6, 6′, and 9 positions [27]. The study [27] is one of the few works in which the properties of a large number of trimethine cyanines of similar structures were systematically studied as potential probes for biomolecules.

It has been shown that substituents at positions 6,6′ increase the tendency of dyes **4**–**13** (Figure 5) to strongly aggregate in aqueous solutions, which hampers the use of these dyes as probes. These dyes show a greater increase in the fluorescence intensity in the presence of DNA than in the presence of RNA, which is favorable for solving tasks of DNA detection [27]. The authors [27] proposed 6,6′-benzoylamino-disubstituted trimethine cyanines as efficient dyes for DNA detection.

Comparing the two similar dyes without 6,6′-substituents studied in [27], *meso*-substituted Cyan 2 (**1**) and *meso*-unsubstituted Cyan 45 (**2**), we can see that the fluorescence growth upon the interaction with DNA for **1** (~177 times) is much greater than for **2** (~4.6 times) due to the very weak intrinsic fluorescence (without DNA) of **1** (mainly present in water as a nonfluorescent *cis*-isomer). Similarly, for other *meso*-substituted dyes studied in [27], intrinsic fluorescence in an aqueous solution is, as a rule, weaker than for the corresponding *meso*-unsubstituted dyes. This imparts to *meso*-substituted dyes the feature of a greater increase in the fluorescence intensity upon the interaction with biomolecules and an advantage for their use as probes.

In [28], the interaction of four carbocyanine dyes, **14**–**17** (Figure 6)**,** with different terminal heterocycles (indo-, thia-, and oxacarbocyanines and a dye with quinoline heterocycles) with DNA was studied using both conventional spectral-fluorescent methods and circular and linear dichroism spectroscopy. Binding to DNA proceeds with both the formation of a complex with the dye in the monomeric form, and the formation of dimeric aggregates in the minor groove of DNA. A slight preference for binding to AT base pairs has been found, and data on intercalation of the dyes upon binding (between base pairs of poly(dG)-poly(dC)) have been obtained. The authors point to a positive correlation between DNA binding constants and the degree of hydrophobicity of cyanine dyes (the partition coefficient in the solvent mixture n-octanol–water).

It has been shown that 3,3′-diethylthiacarbocyanine (**15**, DTC) effectively interacts with DNA to form complexes of two types with binding constants of ~10^6^ and 5 × 10^4^ L mol^−1^. In the complex, nonradiative deactivation processes of the excited state of the dye (vibrational relaxation and photoisomerization) are hindered; in the bound state, 15 exhibits a ~23-fold increase in the quantum yield, reaching ~60% [29]. However, the noticeable quantum yield of the intrinsic fluorescence (2.6% [29]) prevents the widespread use of **15** as a fluorescent probe for DNA.

Molecular docking experiments [30,31] confirm the possibility of the formation of various types of complexes with dsDNA by cationic trimethine cyanine dyes **18**–**21** (Figure 6). Molecular docking can help in studying the types of dye binding to DNA and predict the formation of energetically favorable configurations of carbocyanines in complexes with DNA. The most stable dye complexes (with the most negative ΔG_est_ values) are formed in the minor groove of DNA. The data on docking some *meso*-substituted trimethine cyanines with DNA are presented in Table 1.

Note that the in silico experiment showed the essential role of van der Waals forces and hydrogen bonds in the stabilization of complexes. The possibility of additional stabilization of the dye complex with DNA due to the formation of hydrogen bonds is shown. Docking has shown that the binding of the dyes to DNA can occur in forms close to both *cis*- and *trans*-isomers. The formation of dye complexes in the minor groove of DNA is energetically more favorable than other types of interactions; however, these can also take place in accordance with experimental data.

The possibility of intercalation of the dyes in the complex is confirmed, in particular, by experiments on thermal dissociation of the DNA helix: an increase in the melting point of dsDNA in the presence of dyes **18**, **20**, **21** (Figure 6) was detected [30,31].

The works [32,33] are devoted to the spectral properties of *meso*-substituted trimethine cyanine dye Cyan 2 (**1**; Figure 5) and the study of its interaction with nucleic acids. Cyan 2 was shown to be able to intercalate with dsDNA. The binding proceeds via the formation of an external binding complex (a fast process) which, over time, is rearranged into a partially intercalated complex, which in turn is transformed into a fully intercalated complex. Tests of dye **1** with poly(dA-dT) poly(dA-dT) and poly(dG-dC) poly(dG-dC) show sufficient selectivity of **1** toward GC base pairs. The experiments also show that the dye is prone to self-aggregation (monomer–dimer equilibrium).

Along with *meso*-substituted (symmetrical) trimethine cyanines, unsymmetrical cyanine dyes are also used as nucleic acid probes. A feature of these dyes is very weak fluorescence in an aqueous solution, which can lead to a great relative increase in fluorescence intensity upon interaction with biomolecules. Unsymmetrical cyanine dyes with an acridinium fragment and various polymethine chain lengths (mono-, tri-, and pentamethines) were synthesized, and their binding to dsDNA was studied [34]. The absorption spectra of these dyes range from orange to the near-IR region of the spectrum; in particular, the trimethine cyanine dye **22** (Figure 7) has an absorption maximum at 670–680 nm, which is bathochromically shifted with respect to, e.g., dyes **1**, **2**, and **15** (530–570 nm). It exhibits an increase in fluorescence intensity in viscous solutions and in the presence of dsDNA. The authors believe that the dyes obtained can be used in practice as spectral-fluorescent probes and microviscosity sensors.

A number of unsymmetrical cyanine dyes with a quinoline heterocycle (in particular, trimethine cyanines **23**, **24**; Figure 7) were synthesized, and their properties as spectral-fluorescent probes for dsDNA were evaluated [35]. For a number of the dyes, high Φ_fl_ upon binding to dsDNA (~53–90%) was obtained. It has been shown that the dialkylamino group at position 2 of the quinoline heterocycle suppresses nonradiative relaxation processes of the excited state *S*_1_ in favor of increasing the fluorescence quantum yield of the probe dyes. The developed dyes specifically stain nuclei in fixed HeLa cells in vitro [35].

At the same time, in the work [36], for the unsymmetrical dye **25** (TO-PRO-3; Figure 7), a trimethine derivative of thiazole orange, no fluorescence growth was observed upon binding to DNA. It was shown that the dye forms three different complexes with DNA: two on the surface of the DNA helix and one by intercalation (they have different stoichiometric ratios). Dye 25 was suggested as a deep-red fluorescent indicator with high selectivity to the internal loop structures of the bacterial A-site RNA [37].

Since the spectral-fluorescent and photochemical properties of *cis*- and *trans*-isomers of trimethine cyanine dyes differ significantly [18,19], it is necessary to study the *cis*–*trans* equilibrium in the course of dye interaction with biomolecules. Effects of the interaction with DNA on the *cis*–*trans* equilibrium of a number of *meso*-substituted thia- and oxacarbocyanine dyes, **26**–**32** (see Figure 8) and **1** (Cyan 2; Figure 1), have been studied in a number of works [38,39,40,41]. It has been shown that for the above thiacarbocyanine dyes **26**–**28** and Cyan 2, the interaction with DNA shifts the *cis*–*trans* equilibrium toward the *cis*-isomer, while oxacarbocyanines **29**–**32** bind to DNA mainly in the *trans* form. The authors suggest that this effect is due to steric restrictions for the dye molecule, created by the molecular environment in the complex with DNA. At the same time, for dyes **18** and **19** (Figure 6), when interacting with DNA, there is no complete shift of the *cis*–*trans* equilibrium, and the binding of the dyes to DNA in both *cis*- and *trans*-forms is observed [31].

Oxacarbocyanine analogues of symmetrical thiacarbocyanines were also studied as probes for DNA. In [41], it was shown that a sharp increase in the fluorescence intensity upon binding to DNA is favorable for the use of oxacarbocyanine dyes as spectral-fluorescent probes. For dyes **29** and **30**, the relative increase in the fluorescence intensity at *c*_DNA_ = 2.5 × 10^–4^ mol L^−1^ reaches 55 and 41 times, respectively. The spectral effects observed upon the complexation of oxacarbocyanine dyes are determined by the shift of the isomeric equilibrium toward the formation of *trans*-isomers.

The behavior of *meso*-aryl-substituted cyanines differs from that of *meso*-alkyl-substituted dyes. The spectral-fluorescent and photochemical properties of *meso*-aryl-substituted trimethine cyanines **20** and **21** (Figure 6) in solutions and their interaction with DNA were studied in works [30,42]. The planar structure (and the possibility of torsion) of aryl *meso*-substituents may mean that these bulky substituents do not create barriers to the formation of energetically favorable *trans*-isomers. Therefore, dyes **20** and **21** are in the form of *trans*-isomers both in solutions (regardless of polarity) and in complexes with DNA. This feature and the fact that the interaction of the dyes with DNA occurs involving the aggregated forms of the dyes determine a moderate increase in the fluorescence intensity of dye monomers **20** and **21** in the presence of DNA (by 11.1 and 7.8 times, respectively, at *c*_DNA_ = 5.0 × 10^–4^ mol L^−1^). Molecular docking of the dyes with dsDNA confirms the possibility of the formation of complexes of different types, including: intercalation between base pairs and in the grooves of the double helix of DNA. The possibility of intercalation of the dyes in the complex is confirmed by experiments on thermal dissociation of dsDNA in the presence of the dyes, as well as those on the interaction of the dyes with ssDNA. The limits of detection and quantification of DNA were determined.

The interaction of dyes **18** and **19**, having hydroxyethyl substituents at N atoms of terminal heterocycles, with dsDNA in aqueous buffer solutions leads to the formation of noncovalent complexes, which is accompanied by a significant increase in the fluorescence intensity of the dyes (Φ_fl_~2.3% and 50%, respectively, at *c*_DNA_~5 × 10^–5^ mol L^−1^) [31]. The effect of temperature and NaCl additions on the stability of dye–DNA complexes was studied. The results of experiments on molecular docking show the possibility of several types of binding, both in the groove and intercalation, which is confirmed by the data obtained in the experiments. The effective binding constants of dyes **18** and **19** to DNA were estimated from spectral-fluorescence data (*K*_eff_~1.2 × 10^5^ L mol^−1^ at a stoichiometric ratio dye:DNA = 1:10) and LOD = 0.2 and 0.022 µmol L^−1^, respectively. The DNA detection limit for dye **19** is comparable to that of a conventional nucleic acid stain, ethidium bromide; therefore, this dye has enough sensitivity for use as a DNA probe.

The use of the unique ability of cyanines to form J- and H-aggregates for probing biomolecules was studied in a number of works. Aggregation of *meso*-substituted trimethine cyanine dye **33** (L-21; Figure 9) was studied in binary solutions of DMF–Tris-HCl buffer (pH = 8) containing DNA or RNA. [43]. Binding of dye **33** in the minor groove of DNA leads to the appearance of J-aggregates and dye dimers. The absorption and luminescence bands of J-aggregates of **33** bound to DNA exhibit specific properties that allow the use of dye **33** as a fluorescent probe for DNA.

In [44], J-aggregation of cyanine dyes was used to study conformational changes in various types of nucleic acids, including: ss- and dsDNA, cDNA, and RNA. One monomethine cyanine and three trimethine cyanine dyes (structures **34**–**36**; Figure 9) were studied.

Various modifications of the structures of tri- and pentamethine cyanine dyes (in particular, trimethine cyanines **1, 2, 6**–**8, 33, 37, 38**–**47**; Figure 5, Figure 9 and Figure 10) that are favorable for the use of the dyes as fluorescent probes have been considered [45]. Appropriate modifications in the polymethine chain and/or in heterocycles can lead to a sharp decrease in the fluorescence intensity of the unbound dye, which results in a sharp increase in the fluorescence intensity upon complexation with dsDNA and allows the use of modified tri- and pentamethine dyes as fluorescent probes in detection of dsDNA. In particular, the 6,6′-benzoylamino-disubstituted dyes **6** and **7** demonstrate high (up to 200 times) emission intensity enhancement in the presence of DNA. Due to their selectivity to DNA vs. RNA, these dyes can be considered as potential fluorescent probes for DNA detection [45].

In [46], the fluorescent properties of trimethine cyanine 3,3′-diethyl-4,5,4′,5′-dibenzo-9-methylthiacarbocyanine (**43**, Stains-All) and its structural isomer 3,3′-diethyl-6,7,6′,7′-dibenzo-9-methylthiacarbocyanine (**44**, iso-Stains-All) were determined in the presence of DNA and RNA (Figure 10). The interaction with nucleic acids leads to partial decomposition of aggregates present in aqueous solutions of the dyes and an increase in monomer fluorescence. While iso-Stains-All shows the greatest increase in fluorescence intensity with dsDNA, the greatest increase for Stains-All is observed in the presence of RNA. It was concluded that both dyes can be used for fluorescent detection of nucleic acids.

The synthesis of new unsymmetrical trimethine cyanine benzothiazole-quinolinium dyes (dyes **48, 49**; Figure 11) is reported in [47]. The dyes with high effective binding constants bind to dsDNA and RNA and show selectivity toward A-T DNA sequences. The authors explained the binding selectivity by structural features of the minor groove of the double helix rich in A-T DNA fragments.

In the cycle of works [48,49,50], it has been shown that the trimethine cyanine homodimeric dye BOBO-3 (**50**; Figure 11), which exhibits an increase in the fluorescence intensity upon binding to ds- and ssDNA, forms two types of complexes with DNA: a low affinity, electrostatically driven complex and a full bis-intercalation complex within the DNA double helix. Its interaction with ds- and ss-oligonucleotides of various compositions was also studied. It was shown that **50** forms H-aggregates with single-stranded oligonucleotides. The nucleotide bases C or A directly affect the aggregation of the dye, and the aggregates are highly stable; furthermore, the aggregation of **50** prevents the hybridization of the bases bound with their complementary chains.

The interaction of trimethine cyanine **14**, (Pinacyanol; Figure 6) with nucleic acids was studied in [51]. It was shown that the dye interacts with nucleic acids not by intercalation, but by an outside stacking binding, in which nucleic acids act as templates (H-aggregates are formed). In this case, the interaction of the dye with nucleic acids is not purely electrostatic.

The approach of dye molecules to each other upon the interaction with DNA creates conditions for an excitation energy transfer between donor and acceptor molecules. In [52], the nonradiative energy transfer was studied between two trimethine cyanine dyes, dye **30** (energy donor) and **15** (DTC, acceptor), both of which are noncovalently bound to DNA. The results were discussed from the point of view of the concentrating of the dyes in the microphase (pseudophase) of the biopolymer, creating the conditions for energy transfer.

G-quadruplexes, which are important structures in the field of nucleic acids, also need to be studied. G-quadruplexes are stable four-stranded structures that are capable of forming nucleic acid oligomers. G-quadruplexes are made up of G-quartets, which are planar associates of four guanidine bases held together by eight hydrogen bonds. G-quadruplexes are widely distributed in the telomeric and promoter regions of several important oncogenes and play an important role in cancer cell proliferation [53,54]. They also play an important role in the assembly and regulation of nucleic acid nanostructures. Trimethine cyanine dyes are also used for their study. The processes of aggregation and deaggregation of dimeric trimethine cyanine dyes (thiacarbocyanines) bound to dimers by oligooxyethylene linkers of various lengths (structures of type **51**; Figure 12) were studied. It has been shown that the aggregates formed by **51** in PBS decompose to the initial dimers in the presence of tetramolecular G-quadruplexes [55]. It was concluded that the results provide a clue to the development of highly specific probes for G-quadruplexes.

Controlled aggregation of dye **52** (Figure 12) was studied in the presence of tetramolecular G-quadruplexes (G4-DNA) [56]. It was found that H-aggregates of dye **52** decompose to monomers in the presence of G4-DNA. It was concluded that the dye can serve as a chemosensor for G4-DNA; in particular, it responds to the subtle structure of the conformers of G4-quadruplex.

Aggregation–deaggregation effects of other trimethine cyanine dyes, **53** (DEC) and **54** (Figure 12), were used to detect G-quadruplexes of DNA [57,58]. It is known that a significant number of ligands, when interacting with G-quadruplexes, bind to G4 at their ends (end stacking); rare ligands are capable of binding in the grooves. The cationic *meso*-substituted cyanine dye **54** binds to parallel G-quadruplex both by end stacking and in the grooves (as a monomer or dimer) [57]. In the presence of G-quadruplexes, sharp changes are observed in the absorption and fluorescence spectra of the anionic dye **53** as a result of the decomposition of J-aggregates to form fluorescent monomers (an increase in fluorescence intensity occurs at λ_fl_ = 600 nm by ~70 times), which is not observed in the presence of ds- and ssDNA [58].

It should be emphasized that in several of the works discussed above, the unique property of cyanine dyes that allows them to form ordered J- and H-aggregates is used, which sharply distinguishes these dyes from other types of dyes and expands their use as probes.

Aptamers present another promising application of cyanine dyes. Aptamers are short single-stranded DNA or RNA sequences that are selected in vitro on the basis of their high affinity for the target molecule. Dye-binding aptamers are promising tools for real-time detection of not only DNA or RNA sequences, but also proteins of interest, both in vitro and in vivo. For this purpose, unsymmetrical trimethine cyanine dye **55** (DIR; Figure 12) was synthesized and studied [59]. The dye demonstrates a sharp increase in fluorescence intensity in the presence of the RNA aptamer obtained for this dye (a 60-fold increase upon binding to the RNA aptamer), but weakly fluoresces in solutions and in the presence of dsDNA. In [60] a binary RNA aptamer probe based on trimethine cyanine dye Cy3 (derivative of dye **17**; Figure 6) was developed and studied. When bound to the aptamer, Cy3 fluorescence increased. The aptamer probes obtained are suitable for applications in fluorescence sensing and labeling.

Dye **55** was also proposed as a highly specific fluorescent G-quadruplex probe (in the red region > 650 nm) [61]. Binding to G-quadruplexes results in a 10–70-fold increase in the fluorescence quantum yield of the dye.

The data on the properties of trimethine cyanine dyes that are promising as probes for DNA are summarized in Table 2. The extended version of Table 2 is presented in the Appendix A.

## 3. Influence of Interaction with Proteins and Other Biomolecules on the Spectral-Fluorescent Properties of Trimethine Cyanine Dyes

Protein molecules are very important and the most complex chemical compounds; they are characterized by large sizes and extraordinary diversity, which is created by 20 amino acids that make up their composition and are connected in polypeptide chains in different orders. Proteins play a leading role in life processes precisely because of their extremely diverse composition and structure. Due to the wide variety of types of protein molecules in living organisms, the interaction of cyanine dyes with proteins has not been studied as thoroughly as their interaction with nucleic acids.

Cyanines are currently used for protein staining in gel-based proteomic analysis [62]. Trimethine cyanine dyes of the Lucy family (with the general structure **57**; Figure 13) are used for staining in determining proteins by electrophoresis (see, e.g., [63,64]).

It has been found that a number of cyanine dyes that interact with nucleic acids (in particular, dyes **1**, **2**, **4**, **5**, **10**, **11**, **43**, **44, 46, 47**; Figure 5 and Figure 10) are also capable of noncovalent interaction with serum albumins. In particular, it was found in [65] that trimethine cyanine dyes with *meso* substituents and bridges in the polymethine chain, containing extended heterocyclic systems or N-phenyl and N-cyclohexyl substituents, along with the interaction with nucleic acids, exhibit increased affinity for bovine serum albumin (BSA) in solutions (dyes **46, 58, 59**; Figure 10 and Figure 13). An interaction with BSA with an increase in fluorescence intensity was found for 6,6′-substituted trimethine cyanines **5** and **10** (Φ_fl_ growth by ~22 and ~11 times, respectively) [27].

In [15], the change in the spectral-fluorescent properties of polymethine dyes of various classes upon interaction with human serum albumin (HSA) was studied in detail. Along with other polymethines (anionic oxonols **60**–**62** and tetracyanopolymethines—structure **63**; Figure 14), cationic thiacarbocyanines (dyes **15, 19, 64, 65**) and anionic hydrophilic thiacarbocyanines with sulfonate groups (**53**, **66**–**68**) have been studied. It has been found that binding to HSA leads to a long-wavelength shift in the absorption spectrum and, in most cases, to an increase in the fluorescence intensity of the dyes (in the case of oxonols (**60**–**62**), the increase in fluorescence is very small). The binding constants *K*_eff_ vary from 10^4^ L mol^−1^ (for cationic trimethine cyanines) to (5–6) × 10^7^ L mol^−1^ (for oxonols), whereas the *K*_eff_ value for anionic dyes (**53**, **66**–**68**; Figure 12 and Figure 14) is much higher than for cationic ones. The binding of *meso*-substituted trimethine cyanines leads to *cis*–*trans* isomerization and, as a consequence, a sharp increase in the fluorescence intensity. Aggregates that form the dyes in aqueous solutions decompose upon interaction with HSA, while binding to HSA is often accompanied by the formation of dye aggregates on albumin molecules [15].

Based on the data obtained in the above work, a search was made for an effective spectral-fluorescent probe for albumin. Due to the fact that the anionic trimethine cyanine dye **53** (DEC; Figure 12) upon binding to HSA demonstrates a sharp increase in fluorescence with a high binding constant (*K*_eff_ > 10^6^ L mol^−1^), as well as a strong long-wavelength shift of the absorption spectrum [15]), it was tested in the presence of serum albumins of various animals and in real biological systems containing albumin [66,67,68]. It was found that **53** effectively interacts with HSA, but not with serum albumins of other animals [69]. Dye **53** has also been tested as a probe for the detection of albumin in the human vitreous body and has been shown to have little to no interaction with vitreous components such as hyaluronic acid and alpha fetoprotein [70,71].

Experiments on electronic excitation energy transfer (FRET) provide information on the spatial arrangement of ligand molecules in a complex [72,73]. Thus, in [73], FRET experiments were carried out, in which anionic trimethine cyanine dyes (dyes **53, 66**–**68**) served as energy acceptors, and tetracyanopolymethine **63** served as a donor, with the donor and the acceptor being in a complex with HSA. Energy transfer efficiencies, critical Förster radii, and distances between the donor and the acceptor were determined.

The spectral-fluorescent properties of the anionic oxacarbocyanine dye **69** (OCC; Figure 14) were studied in solutions and in complexes with HSA and BSA [74,75,76]. Interaction with albumins leads to a shift in the *cis*–*trans* isomeric equilibrium of **69** and to a significant increase in dye fluorescence (Φ_fl_~57% at *c*_BSA_ = 2.5 × 10^–5^ mol L^−1^). Fluorescence quenching of **69** in a complex with HSA by ibuprofen and warfarin was studied. The data on fluorescence quenching by ibuprofen indicate the binding of the dye to site II of subdomain IIIA in the HSA molecule (Figure 15).

The synchronous fluorescence spectra of HSA in the presence of **69** (Figure 14) have shown that complex formation with **69** does not lead to a noticeable rearrangement of the protein molecule at the binding site. The effect of BSA denaturation (by urea addition) on the spectral-fluorescent properties of the dye in the complex was studied [75]. Violation of the protein structure upon denaturation led to the decomposition of the OCC–BSA complex; it was concluded that the dye was bound to BSA domain III. Denaturation of HSA leads to the formation of J-aggregates of **69** of different types. It was also concluded that **69** could act as an albumin probe and could be used as a test for protein denaturation.

The rational design of spectral-fluorescent probes requires studying the correlation between the efficiency of dye binding with protein, dye aggregation in solutions and in the presence of biomolecules, and the balance of hydrophilic and hydrophobic properties of dyes. The work [77] studied the effect of the number of sulfonate groups (from one to four) in the molecules of four anionic unsymmetrical trimethine cyanine dyes (structures **70**–**73**; Figure 16) on the aggregation properties of the dyes and on their ability to bind to serum albumin (BSA). The selected cyanine dyes were characterized by emission in the near-IR region (about 700 nm). With an increase in the number of sulfonate groups, both dimerization of the dyes and the binding constant with albumin decrease (maximum *K*_eff_~3 × 10^5^ L mol^−1^ for **70** and minimum *K*_eff_~1.3 × 10^4^ L mol^−1^ for **73**). The results can be used for screening of promising fluorescent probes for biological systems [77].

In [8], a series of unsymmetrical trimethine quinoline cyanine dyes with an indole core (structures **74**–**77**; Figure 16) were synthesized and studied. A moderate increase in fluorescence intensity upon interaction with BSA was observed for some dyes (for **76**, by 3.5 times).

The influence of interaction with HSA on the spectral-fluorescent properties of a number of *meso*-substituted anionic trimethine cyanines with sulfonate groups was studied in [78] for **52** (DMC) and **78** (Figure 17), compared to the previously studied dyes **53** (DEC) and **69** (OCC). In aqueous solutions, dye **52** forms, along with dimers and H- and J-aggregates. The thermodynamic parameters of the monomer–dimer equilibrium and the value of the dimerization constant *K*_dim_ for **52** were determined (Δ*H* = −118.7 kJ mol^−1^, Δ*S* = −121 J mol^−1^, Δ*G* = −83 kJ mol^−1^, *K*_dim_~2.1 × 10^7^ L mol^−1^ at 23 °C). The noncovalent interaction of **52** and **53** with HSA leads to the decomposition of dimers with a shift of the *cis*–*trans* isomeric equilibrium toward the *trans*-monomer complexed with HSA, which is accompanied by a significant increase in dye fluorescence (at *c*_HSA_ = 2.86 × 10^–6^ mol L^−1^ Φ_fl_ = 27%). The effect of structural rearrangements of HSA upon urea denaturation on the spectral-fluorescent properties of the dyes has been studied. Binding constants (*K*_eff_) of the dyes to HSA, as well as LOD and LOQ values, were estimated from spectral-fluorescence data. The low LOD and LOQ values (<3.0 × 10^–9^ mol L^−1^) upon binding **52** to HSA, as well as the high *K*_eff_ value (>5.0 × 10^6^ L mol^−1^), allowed the authors to propose dye **52** as a new spectral-fluorescent probe for HSA [78].

A comparative study of the interaction of *meso*-alkyl-substituted anionic trimethine cyanine dyes **79** (OXEC), **80** (DMEC; Figure 17), and **52** (DMC) with HSA and BSA was performed in [79]. Interaction with albumins leads to the formation of complexes of *trans*-isomers of the dyes; however, in the case of BSA, the shift of the *cis*–*trans* isomeric equilibrium is incomplete. The results of molecular docking are consistent with the data obtained from the spectra. Based on fluorescence data, the effective binding constants of *trans*-isomers to BSA (*K*_eff_) and the limits of detection of albumin molecules (LOD and LOQ) were determined. The data obtained indicate a greater selectivity of dyes **52** and **80** toward HSA compared to BSA, while **79** does not show such selectivity.

The experiments on molecular docking of trimethine cyanine dyes with albumins illustrate well the experimental results obtained in vitro. Table 3 shows the results of blind docking of dye–HSA systems using the DockThor server [80,81]. The *cis*-conformations were taken as the initial ones for all the dyes.

The docking has shown that the *trans* configurations of trimethine cyanines are favorable for binding with albumin. Dyes **79** and **69** are also able to interact with the protein in the form of *cis*-isomers. Molecular docking of dye–HSA systems was carried out; the results correspond to the binding of **52** on the surface of domain I of HSA (dye **53** is probably able to bind to domain IIA of HSA) [78,79].

Binding to HSA leads to the induction of a spectrum of circular dichroism (due to violation of the symmetry of dye molecules in the complex with albumin) and, in some cases, is accompanied by aggregation of dyes on albumin molecules. These aggregates often give biphasic circular dichroism spectra. At the same time, aggregates formed by the dyes in the absence of HSA (for example, **63**, **65**, and **67**) decompose in its presence [15]. The behavior of three hydrophilic *meso*-substituted carbocyanine dyes, **52**, **53**, and **81** (Figure 18), in the presence of HSA was studied by spectral-fluorescence and circular dichroism spectroscopy [82,83]. The monomeric dyes are in equilibrium with J-aggregates in solutions; in the presence of HSA, the equilibrium is shifted toward monomeric cyanine molecules. It has also been shown that HSA induces the chiral properties of dye J-aggregates, which is reflected in the bisignate CD signal of J-aggregates. It was concluded that the *meso* substituent plays an important role in the interaction of HSA with J-aggregates.

The influence of protein–ligand interaction on the spectral-fluorescent properties of trimethine cyanine dyes was also studied using both UV–Vis spectroscopy and near-infrared laser-induced fluorescence (NIR-LIF). Nine dyes were studied: three indocarbocyanines (structures **82–84**), three benz[e]indocarbocyanines (**85–87**), and three benz[c,d]indocarbocyanines (**88–90**; see Figure 18) [84]. It has been shown that the efficiency of interaction with HSA is affected not only by the overall hydrophobic properties of the molecules, but also by the size of the ligands. The replacement of the ethyl groups in the indole side chains (dye **82**) by butyl groups (dye **83**) leads to an improvement in the binding characteristics and an almost threefold increase in the affinity constant with HSA. The introduction of phenylpropyl groups (dye **84**) instead of the ethyl groups improves the compatibility of the dye with the HSA binding sites, but causes steric hindrance for interaction.

Some data on the properties of trimethine cyanine dyes that are promising as probes for BSA and HSA are summarized in Table 4. More data can be found in the Appendix A.

A relatively small number of works have been devoted to the study of the noncovalent interaction of polymethine dyes with proteins other than albumins. The effect of a number of proteins—including ribonuclease A (RNase), lysozyme, trypsin, and BSA—on J-aggregation of three *meso*-substituted thiacarbocyanine dyes, **91**, **53**, and **80** (Figure 12, Figure 17 and Figure 19), in solutions was studied in detail [85]. The monomer–dimer–J-aggregate equilibrium was studied; the formation of J-aggregates correlates with the decomposition of dye dimers. RNase stimulates J-aggregation of all the three dyes, whereas lysozyme stimulates **91** and **53** (DEC), and trypsin stimulates only **53**. At the same time, interaction with BSA leads to deaggregation of dimers of the dyes and the formation of highly fluorescent complexes of dye *trans*-monomers with albumin (for example, Φ_fl_ = 22% for the complex **91**–BSA) [85].

It was shown in [86] that gelatin stimulates the formation of J-aggregates of some anionic trimethine thiacyanine dyes with sulfonate groups (in particular, **52** and **53**); it was also shown that aggregates are formed from dye dimers. The kinetics of the formation of J-aggregates of **53**, **69**, and **91** in the presence of gelatin were studied in [87] and in the presence of gelatin, lysozyme, and trypsin in [88].

The influence of collagens on the spectral properties of polymethine dyes of various classes was studied in [16]. It has been shown that collagens stimulate the formation of J-aggregates of anionic thia- and oxa-trimethine cyanine dyes with sulfonate groups, which themselves can form J-aggregates in aqueous solutions, namely **53** and **69** (for dyes **15**, **60**, **80** in the presence of collagen, a weak formation of disordered aggregates was observed, which reduced the intensity of the absorption band of the initial dye). It was shown that the spectral properties of J-aggregates of **53** in the presence of collagen differ from those in the presence of gelatin or without additives; they also differ for different types of collagens. This feature of dye **53** made it possible to use this dye as a probe for collagens (and even to assess the possible type of collagen) in extracellular media of organisms, particularly in the aqueous humor and vitreous body of a frog [68].

J-aggregation of the anionic oxacarbocyanine **69** in aqueous solutions in the presence of various proteins and polyelectrolytes was studied [76]. It has been shown that fibrillar proteins (along with collagens, immunoglobulin G) and polyelectrolytes stimulate J-aggregation of **69**.

The chiral properties of J-aggregates of trimethine cyanine **69** formed in the presence of biomolecules (in particular, lysozyme, trypsin, ribonuclease, gelatin, DNA) were studied using circular dichroism (CD) spectroscopy [89]. It has been shown that the chirality of the resulting dye J-aggregates is controlled by the chirality of biomolecules.

There are a number of works devoted to the search for probes for amyloid fibrillar proteins, due to their association with some neurodegenerative diseases (in particular, Alzheimer’s and Parkinson’s diseases). In [90], a number of mono-, tri-, penta-, and heptamethine cyanine dyes with benzothiazole and benzimidazole heterocycles, as well as squarylium dyes, were studied in order to develop probes for the detection of fibrillar beta-lactoglobulin (FβLG). The properties of trimethine cyanines **2**, **13**, **18**, and **92** (see Figure 19) were studied. It was found that *meso*-substituted cyanine dyes with amino groups in heterocycles (including trimethine cyanines **13** and **18**) are able to preferentially bind to FβLG and demonstrate an increase in fluorescence intensity in the presence of the protein.

A search for dye probes for the fluorescent detection and quantitation of fibrillar α-synuclein (ASN) was carried out in [91,92]. Mono- and trimethine cyanine dyes (trimethine cyanines **13** and **92**) with terminal benzothiazole, pyridine, and quinoline heterocyclic groups have been studied. The formation and accumulation of ASN amyloid fibrils in the brain is a key feature of Parkinson’s disease. The incorporation of amino or diethylamino substituents at position 6 of the cyanine benzothiazole heterocycle results in a selective fluorescence response to the presence of fibrillar ASN. It was also found that the *meso*-substituted trimethine cyanine **92** exhibits a higher fluorescence intensity and selectivity for aggregated ASN than the conventional amyloid dye probe Thioflavin T [91]. Using fluorescence spectroscopy and atomic force microscopy, it was shown that trimethine cyanine dye **92** is a sensitive fluorescent probe for fibrillar ASN in vitro. The dye demonstrates good reproducibility and allows selective recognition of amyloid proteins of different amino acid compositions [92]. Trimethine cyanine dye **93** (Figure 20) was proposed in [93] as a sensitive spectral probe for monitoring the transition of monomeric insulin to the fibrillar form and for its use in the insulin aggregation inhibition assays in vitro. The influence of various N,N’-substituents in the molecules of benzothiazole trimethine cyanine dyes (dyes **94**–**103**; Figure 20) on their ability to sense protein amyloid aggregates was studied [94]. The dyes with butyl, hydroxyalkyl, and phenylalkyl substituents show increased sensitivity to fibrillar lysozyme, while the dyes with quaternary amino groups showed sensitivity to fibrillar insulin.

A series of 18 new cyanine dyes (mono-, tri-, penta-, and heptamethine cyanines, including unsymmetrical cationic trimethine cyanines **104–109**; Figure 20) were studied in [95]. The potential of new cyanine dyes to inhibit the formation of amyloid structures in insulin was evaluated. According to their ability to inhibit the formation of amyloid fibrils, the dyes are arranged in the order tri- > penta- > mono- > heptamethine cyanines. Trimethine cyanines **105** and **106** almost completely prevented protein aggregation. Molecular dynamics modeling showed an increase in insulin helicity in the presence of cyanines. Two mechanisms of inhibition of insulin fibrillation are proposed: stabilization of the native protein structure followed by the retardation of the protein nucleation (all dyes), and blocking the lateral extension of β-sheets via the dye–protein stacking interactions.

The effects of enhancing the fluorescence signal of amyloid-sensitive trimethine cyanine dyes by silver island films were studied in [96]. Benzothiazole dyes **99** and **102** (Figure 20) were chosen for the study. In the case of zwitterionic dye **102**, the presence of silver islands increased the fluorescence intensity by factors of 5.2, 6, and 3.4 for the free dye and its complexes with insulin and lysozyme fibrils, respectively. The data obtained create the basis for increasing the sensitivity of probes for the detection of protein amyloid fibrils.

The nonradiative electronic excitation energy transfer in amyloid fibrils between thioflavin T (electronic energy donor) and unsymmetrical trimethine cyanine dyes **104–108** (acceptors) was studied [97]. The transfer of electronic excitation energy occurs via the resonance mechanism (FRET). It was concluded that FRET between Thioflavin T and cyanines could be employed for amyloid detection.

The mechanisms of interaction between the anionic *meso*-substituted trimethine cyanine **52** and human transferrin, as well as conformational changes in the protein, were studied in [98]. Transferrin (hTf) is a protein whose function is to control the amount of free iron in the blood plasma. In a complex with hTf, the dye nonradiatively quenches protein fluorescence (static quenching); the quenching proceeds with a high constant of 10^9^ L mol^−1^. Binding of dye **52** occurs in the N-lobe of the protein and leads to an increase in the content of α-helices and increased hydrophobicity around tryptophan residues of hTf. Spectral-fluorescent and CD spectroscopies as well as molecular modeling have been used to study the binding mechanisms of dye **53** to hTf at various pH and temperatures [99]. As well as being structurally similar to **52**, trimethine cyanine **53** binds to hTf in the N-lobe with a high *K*_eff_~10^7^  L mol^−1^; quenching of intrinsic fluorescence of hTf by the dye is observed. The binding of dyes **52** and **53** to hTf causes an increase in the content of α-helices of the biomolecule. A new method for determining hTf conformation using J-aggregation of **52** was proposed in [100]. The open form of hTf (apo-hTf) strongly stimulates the aggregation of this dye, while the closed form (holo-hTf) hardly does. This allows detection of sub-micromolar levels of apo-hTf against a background of holo-hTf. In [101], hTf was determined using the effects of aggregation of dye **53**. It was shown that apo-hTf induces the transition of dimeric aggregates **53** to long-wavelength J-aggregates. Holo-hTf does not have this pronounced ability. LOQ for apo-hTf~8–80 nM (LOD~2.8 nM) was obtained; the interaction is characterized by a high binding constant (~10^6^ L mol^−1^) and selectivity [101].

On the basis of trimethine cyanine dye BOBO-3 (dye **50**), a new aptamer fluorescent probe beacon was developed [102]. Its efficiency for the detection of thrombin was shown. When a thrombin sample is mixed with the probe, BOBO-3 is competed away from the beacon due to target-induced aptamer folding, which then causes a decrease in the BOBO-3 emission mediated by energy transfer (FRET) from quantum dots. High specificity at the nanomolar limit of sensitivity was achieved.

The noncovalent interaction of cationic trimethine cyanines **1**, **2**, **15**, and **110** (Figure 21) with hyaluronic acid (HA), a glycosaminoglycan and one of the most important biopolymers in a living organism, was studied [103]. In the presence of HA, the dyes form H-aggregates, and according to their tendency to aggregation, the dyes are arranged in the series **1** > **2** > **110** > **15**. For dye **1**, the aggregation number *n* = 3 was determined; the dye was proposed as a spectral probe for detecting HA in biological systems. The noncovalent interaction of **1** with another glycosaminoglycan—chondroitin-4-sulfate (C4S, which is incorporated into cartilage, tendons, and synovial fluid of animal joints)—in buffer solutions with different pHs and in water in the absence of buffers was studied using the spectral-fluorescent method [104]. It was shown that, under all conditions studied, at relatively high concentrations, dye **1** binds to C4S predominantly as a monomer, which is accompanied by a sharp growth of fluorescence (intermediate formation of dye aggregates on C4S is also observed). The dependence of the fluorescence quantum yield (and the binding constant) on pH is nonmonotonic: it is minimum at neutral pH and rises in the acidic and basic pH regions. This can be explained by pH-dependent changes in the charge of the C4S macromolecule and its related conformational changes, which can affect the rigidity of the dye molecule and the energy of its interaction with the macromolecule. Since the changes in the absorption spectra of the dye and the growth of its fluorescence are observed even at low concentrations of C4S, dye **1** can be suggested as a probe for detecting C4S.

The interaction of trimethine cyanine dyes **53** (DEC) and **1** (Cyan 2) with biological surfactants—including bile salts sodium cholate, sodium deoxycholate, and sodium taurocholate, as well as, for comparison, the synthetic surfactant analogue sodium dodecyl sulfate (SDS)—was studied using spectral-fluorescent methods in a wide range of surfactant concentrations [105]. For **53**, decomposition of dye dimers into *cis*-monomers and *cis–trans* conversion of the monomers formed (with fluorescence growing) are observed. Upon the introduction of increasing concentrations of bile salts, decomposition of dye dimers into the monomers begins at lower concentrations than *cis–trans* conversion. The former process is almost completed at concentrations close to the CMC of secondary micelles of bile salts (CMC_2_), while the latter process occurs even at bile salt concentrations much higher than CMC_2_. Hence, dye **53** can serve as a probe that allows for estimating the value of CMC_2_ and is indicative of reorganization of secondary micelles upon an increase in bile salt concentration.

## 4. Influence of Interaction with Biomolecules on Photoisomerization of Trimethine Cyanine Dyes

The noncovalent interaction of cyanine dyes with biomolecules, as a rule, hinders dissipation of excitation energy over the intramolecular degrees of freedom of trimetine cyanine dye molecules and leads to a drop in the quantum yield of photoisomerization (accompanied by an increase in the quantum yield of fluorescence as a competing process) [12]. A decrease in the photoisomer signal was observed upon flash photolysis of *meso*-unsubstituted dye **15** (DTC) in the presence of dsDNA [29].

Upon flash photoexcitation of solutions of *meso*-substituted trimethine cyanine dyes (in particular, dyes **19** and **53**) in polar solvents, in which they are mainly in the form of the *cis*-isomer, photoisomerization is usually not observed, but it is observed in low-polarity solvents, in which they are in the form of the *trans-*isomers [18] (cyanine **27** is an exception, because for this dye, photoisomerization of both *trans*- and *cis*-isomers was observed [106]). For *meso*-substituted trimethine cyanines **18** and **19**, photoisomerization was also observed in a nonpolar solvent, dioxane, but it was not observed in polar solvents. Photoisomerization was not observed in complexes with dsDNA either [31]. Photoisomerization in a complex with dsDNA was also not observed for *meso*-substituted trimethine cyanine **26** [38].

*Meso*-aryl-substituted trimethine cyanines **20** and **21** occur in all media as *trans*-isomers, since planar *meso*-aryl substituents do not create significant steric hindrances for more stable *trans*-isomers. Upon flash photolysis of solutions of these dyes, both in polar and nonpolar solvents, the formation of photoisomers (in the *cis-*form) is observed (Figure 22); in aqueous phosphate buffer, the yield of photoisomers decreases (by a factor of ~20), and when dsDNA is introduced (2–4 × 10^–4^ mol L^−1^), photoisomerization is not observed at all [42]. Photoisomerization of these dyes is also suppressed in complexes with ssDNA [30].

Interaction with DNA also hinders photoisomerization processes in molecules of oxacarbocyanine dyes **16**, **29,** and **30**. Upon flash photoexcitation of aerated solutions of dye **16** in the presence of dsDNA (*c*_DNA_ = 2.4 × 10^–4^ mol L^−1^), the intensity of the absorption signal of the photoisomer decreased, and its lifetime increased. For *meso*-substituted oxa-dyes **29** and **30** at *c*_DNA_ = 4.4 × 10^–5^ mol L^−1^ and higher, no photoisomer signals were observed [107].

Photoisomerization was not observed upon flash photolysis of anionic *meso*-substituted trimethine cyanines **53**, **52**, **79**, and **80** in complexes with HSA and BSA either [108].

However, binding to biomolecules can suppress as well as stimulate photoisomerization. The photochemistry of *meso*-substituted trimethine cyanines **1** (Cyan 2) and **53** (DEC) was studied in the presence of micelles of bile salts (natural surfactants sodium cholate, sodium deoxycholate, and sodium taurocholate) in [109]. Upon flash photolysis of dyes **1** and **53** in water in the presence of micelles of bile salts, signals of photoisomers were observed, although these signals were absent in aqueous solutions without surfactants. This is due to the conversion of nonphotoisomerizable *cis*-isomers of the dyes in aqueous solutions to the photoisomerizable *trans*-form upon binding to micelles. The lifetimes of photoisomers of the dyes (dark back isomerization to the initial *trans*-isomers) were 60–190 μs.

Stimulation of photoisomerization of *meso*-methyl-substituted trimethine cyanines **1** (Cyan 2) and **110** upon noncovalent binding to dsDNA and chondroitin-4-sulfate was found in [110] Signals corresponding to the formation and dark decay of *trans*-photoisomers of the dyes in complexes with biomolecules (in complexes they were in the *cis* form) were detected by flash photolysis, although in the absence of biomolecules, there were no signals of photoisomers (for **1**, the photoisomer lifetime was τ = 3.9 and 2.2 ms in the presence of DNA and C4S, respectively). To explain the effect of stimulation of photoisomerization of *cis*-isomers, it was assumed that the biopolymer matrix affects the potential photoisomerization surfaces of the dyes: when interacting with the biopolymer, the potential surface of the *S*_1_ level is distorted with a shift of the minimum toward the *trans*-isomer [110].

## 5. Influence of Interaction with Biomolecules on Generation and Decay of the Triplet State of Trimethine Cyanine Dyes

As a rule, the quantum yield of the triplet (T) state for trimethine cyanines in solution is very low (<0.05) [11]. Noncovalent interaction with biomolecules usually hinders nonradiative channels of deactivation of the excited singlet state of the dye by intramolecular motions, which enhances fluorescence emission and intersystem crossing to the T state as competitive processes and allows detection of T–T absorption signals upon flash photolysis.

In particular, in [111], when studying the noncovalent interaction of dye **15** with dsDNA, an increase in the quantum yield of intersystem crossing of **15** to the T state in a complex with DNA was found. The decay kinetics of the T state of the dye in the presence of dsDNA were biexponential, which corresponded to the T states of two types of dye–DNA complexes, probably surface (in the groove) and intercalation complexes, which had different spectral and kinetic characteristics.

Data on the increase in intersystem crossing to the T state in complexes with DNA were obtained for a large number of *meso*-substituted trimethine cyanines, particularly **1**, **27**, **18**, **19**, **20**, **21**, **26**, **28**, and **110** [31,38,42,106,110,112] (Figure 23). Intersystem crossing to the T state in complexes with DNA also increased for oxacarbocyanines **16**, **29**, and **30** [107]. The decay kinetics of the T states in all these cases were biexponential. In particular, for dye **20**, two components of the decay kinetics had rate constants of *k*_1_~1.2 × 10^4^ s^−1^, *k*_2_~1.2 × 10^3^ s^−1^ (*c*_dsDNA_ = 4 × 10^–4^ mol L^−1^); for **21**, these were *k*_1_~1.5 × 10^4^ s^−1^, *k*_2_~2.0 × 10^3^ s^−1^ (*c*_dsDNA_ = 4.2 × 10^–4^ mol L^−1^). At the same time, the contribution of the long-lived component to the kinetics is much less than that of the short-lived one, which may be due to the binding of the dye on the surface of the DNA helix [42].

It was found that an increase in the intersystem crossing of cyanine dyes to the T state is caused by the interaction not only with dsDNA, but also with ssDNA. In complexes of dyes **20** and **21** with ssDNA (2–8 × 10^–4^ mol L^−1^), triplet quantum yields were 5.4% and 0.7%, respectively. In this case, the decay kinetics of the T states of the dyes were monoexponential, which indicates the formation of only one type of dye–ssDNA complex (Figure 6) [30].

Primary photochemical processes (including the formation of a triplet state) involving *meso*-substituted anionic trimethine cyanine dyes **53**, **52**, **79**, and **80** in complexes with HSA and BSA were observed [108]. In the presence of serum albumins (in the absence of oxygen), the appearance of T–T absorption signals was detected. The average lifetimes of T states of the dyes in complexes with albumins were in the range of 0.35–1.2 ms. The decay kinetics of the T states in some cases deviate from monoexponential, which can be explained by simultaneous binding of *trans*- and *cis*-isomers of the dyes to albumin and, consequently, by superposition of the decay kinetics of the T states of both isomers upon registration. For **52**, **53,** and **80** with HSA, the T state decay kinetics are described by single-exponential dependences, since the dyes form a complex with HSA only as *trans*-isomers.

The generation of a T state was also observed upon flash photolysis of *meso*-substituted cyanines **1** and **110** in complexes with chondroitin-4-sulfate [110].

It was shown that the T states of the dyes are generated upon flash photolysis of trimethine cyanines **1** and **53** in micellar systems of bile salts (sodium cholate, deoxycholate, taurocholate) and sodium dodecyl sulfate (SDS; taken for comparison) [109]. It was concluded that the photochemical behavior of the dyes in micellar systems of bile salts and SDS is similar.

Quenching of the triplet state of a number of *meso*-substituted thiacarbocyanine dyes in complexes with DNA by nitroxyl radicals, iodide ions, and oxygen was studied in [112]. In the presence of DNA, iodide ions did not quench the T states of dye molecules (probably due to the electrostatic repulsion of the quencher anion from the polyanionic DNA molecule). At the same time, the stable nitroxyl radical (4-hydroxy-TEMPO) quenched only long-lived components of the decay kinetics of triplet dye molecules. This supports the hypothesis that the long-lived components of the T-state decay kinetics of cyanine dyes complexed with DNA belong to dye molecules bound in the minor groove of the DNA helix (a surface-type complex), while the short-lived components correspond to intercalation complexes between DNA base pairs.

Quenching of the T states of trimethine cyanine dyes by dissolved oxygen (O_2_) in complexes with DNA was studied in a number of works [31,38,42,106,111,112]. It occurs much more slowly than in the absence of DNA (with sensitization of triplet states of the dyes in the solution). When T states of the dyes are quenched by O_2_ by the mechanism of energy transfer to form singlet oxygen (^1^O_2_), the diffusion-controlled quenching rate constant *k*_q_(O_2_) includes the spin-statistical factor 1/9, and in water one can expect *k*_q_(O_2_) = 1/9 *k*_dif_~3 × 10^9^ mol^−1^ L s^−1^. Indeed, *k*_q_(O_2_) was found to be 1.2 × 10^9^ mol^−1^ L s^−1^ upon oxygen quenching of dye **15** T state sensitized by sodium anthracene-2-sulfonate in water [111].

The values of *k*_q_(O_2_) in complexes with ds- and ssDNA were found to be significantly lower than 1/9 *k*_dif_, which indicates significant steric hindrances to the quenching process in the complexes. For example, in a complex with dsDNA of dye **15** (DTC), *k*_q_(O_2_) = 1.5 × 10^8^ and 4.5 × 10^8^ mol^−1^ L s^−1^ were found for the short-lived and long-lived T components, respectively [111]; for dye **20,** this was *k*_q_(O_2_) = 4.3 × 10^8^ mol^−1^ L s^−1^, and for **21**, this was *k*_q_(O_2_) = 6.0 × 10^8^ mol^−1^ L s^−1^ [42]. In the case of ssDNA for **20** *k*_q_(O_2_) = 3.0 × 10^8^ mol^−1^ L s^−1^ [30].

The *k*_q_(O_2_) values for the T states of dyes **53**, **52**, **79**, and **80** in noncovalent complexes with serum albumins were determined to be *k*_q_(O_2_)~1.3–2.5 × 10^8^ mol^−1^ L s^−1^ [108], which, as in the case of DNA, is much lower than the value of *k*_dif_ due to shielding of the dye molecule in the complex.

## 6. Practical Application of Trimethine Cyanine Dyes in Biophysics and Biochemistry

Due to the wide fields of application of cyanines in biophysical and biochemical studies, this section will consider only some typical works on the biophysical and biochemical applications of trimethine cyanines based on their spectral-fluorescent and photochemical properties. Applications of fluorescent cyanines in biophysical research are widely considered in review [113] and, partially, in review [12].

A number of cyanine dyes were used in gel electrophoresis as fluorescent stains for the detection of DNA fragments [114]. Twenty-three thia-, oxa, and indocyanine dyes (including trimethine cyanine dyes **17**, **111–118**; Figure 6 and Figure 24) with both symmetrical and unsymmetrical structures were studied. Four trimethine cyanine dyes (**17**, **112**, **114**, **116**) were found to be more sensitive than ethidium bromide, which is commonly used for DNA detection.

Various trimethine cyanine dyes are used in practice for visualization of DNA for fluorescence microscopy of biological samples (see Figure 25, structures **119-122**). The work [115] describes a protocol for the use of cyanine dyes (in particular, **25**, **50**, **121**) for DNA visualization in combination with restriction enzymes. The advantage of the proposed method lies in its simplicity and the fact that no laboratory equipment is required.

Trimethine cyanines **23** and **24** (Figure 7) specifically stain nuclei in fixed HeLa cells with confocal laser-scanning microscopy [35]. For the detection of proteins (polyamino acids), a number of trimethine cyanine dyes were developed and patented [116].

The use of monomeric and homodimeric cyanine dyes for the detection of dual-color fluorescence in situ hybridization (FISH) using confocal laser scanning microscopy (CLSM) was studied in [117]. Trimethine cyanine dye **25** (TO-PRO-3; Figure 7) was shown to exhibit nuclear specific staining without any cytoplasmic staining and stability. High stability of the fluorescence signal was also obtained in the case of trimethine cyanine dyes **119** (YO-PRO-3) and **120** (YOYO-3).

As mentioned above, dye **25** can be used as a deep-red fluorescent indicator with high selectivity to the internal loop structures of the bacterial A-site RNA [37]. The dye is able to bind to bacterial RNA loops (A-site) with increasing fluorescence, which makes it possible to control binding to the A-site of other compounds due to their competition with the probe (fluorescent intercalator displacement, FID).

Trimethine cyanine dye **55** (DIR; Figure 12) was proposed for practical use as a highly specific fluorescent G-quadruplex probe in the red region >650 nm, especially for the parallel G-quadruplex *c-myc* [61].

Dyes **25**, **120**, and **121** (TOTO-3) were used to detect cells in mineral soil [118]. The use of dyes **120** and **121** made it possible to detect microbial cells despite a strong background from nonspecifically bound probes.

Cyanine dyes have also found their application for protein staining in gel-based proteomic analysis [61] A number of spectral-fluorescent probes based on trimethine cyanine dyes have been developed and patented for the detection of proteins [63,64,116].

Trimethine cyanines are proposed for use in practice as markers of insulin amyloid structures [119].

Trimethine cyanine **53** (Figure 12) was widely used in the practice of biological research as a spectral-fluorescent probe for the detection and quantification of albumin and collagen in the extracellular media of the developing human eye and for the detection of collagen in the media of frog eye [66,67,68,70,71]. Studies using dye **53** to check the native state of HSA that formed coatings on magnetic nanoparticles showed that some of the HSA biomolecules retained the ability to bind the fluorescent probe, that is, retained native functional properties [72,120]. Dye **53** was also used in the study of the structure and evolution of micelles of biological surfactants—bile salts [105].

The use of trimethine cyanines in combination with aptamers, short single-stranded DNA or RNA fragments with high affinity for target molecules, has already been noted above. Cyanines are also used in single-chain variable fragment (scFv) antibody protein systems to form a protein–dye fluoromodule. In particular, trimethine cyanine **55** (Figure 12) is used for this purpose [121,122]. The introduction of an electron-withdrawing CN substituent into its polymethine chain (to α-position) increases its resistance to photo-oxidative degradation (dye **122**) [123].

Cyanine dyes are also applied in super-resolution fluorescence microscopy based on single-molecule tracking, which allows diffraction limits to be overcome and images with high spatial resolution (STORM, etc.) to be obtained [124,125]. In particular, trimethine cyanine Cy3 (derivative of dye **17**) can be used as an activator for pentamethine cyanine Cy5 in such systems [126,127].

One of the most important applications of cyanine dyes is photodynamic therapy (PDT)—treatment with photosensitizers excited by light. In the PDT protocols of oncological diseases, penta- and heptamethine cyanine dyes, which have long-wavelength absorption maxima, are mainly applied [128,129]. Photoprocesses involving such dyes lead to the generation of reactive oxygen species that cause damage to tumor DNA [130]. More details about the use of cyanine dyes in PDT can be found in reviews [131,132,133].

However, trimethine cyanine dyes are also being studied in this respect. In particular, in [134], the activity of dye **56** (Figure 12) against cancer cells was studied. Dye **56** has the ability to target RNA and accumulates at the site of the tumor, targeting cancer cells (containing a lot of RNA).

The review [135] considers various aspects of practical use of dyes, including cyanine dyes, in medical oncological practice, histopathology, and molecular diagnostics of oncological diseases. The same agent may have dual functionality in the detection and treatment of cancer in a relatively new field known as theranostics. This is facilitated by new generation dyes conjugated to tumor-targeting probes such as antibodies, and these bioconjugated agents may also include nanoparticles or radioisotopes.

## 7. Concluding Remarks

An ideal probe for biomolecules should have a wide range of properties, which is sometimes difficult to achieve in one molecule. First of all, a high binding constant to the target biomolecule is required. For fluorescent probes, the low fluorescence quantum yield in the free state and the high quantum yield in the bound state, that is, a sharp increase in fluorescence intensity upon binding to a biomolecule, are important. For absorption probes, a high extinction coefficient (narrow absorption band) and a significant spectral shift upon binding to a biomolecule are important. In addition, when used in real biological systems, the probe should not interact with other components of the system (including those with a similar chemical nature), i.e., it should have selectivity (for example, when staining DNA in cellular media, it should not interact with RNA present there and vice versa). If the probe dye is used as a monomer, then its use may be hindered by self-aggregation in aqueous media, which should be minimized. At the same time, the unique ability of cyanines to form ordered J- and H-aggregates (by themselves or on a biomacromolecule as a template) is sometimes used in the design of probes, and in this case the probe should have this ability. For the quantitative detection of biomolecules, the calibration curve is also necessary, which should be linear in a wide range of concentrations, and to ensure high sensitivity, low detection limits (small LOD/LOQ values) must be achieved. From a practical point of view, sufficient solubility of a probe in water is necessary, as well as photo- and dark stability, as it enables researchers to work with it for a sufficient period of time. All these probe properties should be provided by its peculiar molecular structure. For example, the positive charge of a probe molecule increases its efficiency (binding constant) of interaction with DNA; intercalation between DNA base pairs is characteristic of cationic probes with planar aromatic rings, whereas binding in the minor groove requires flexibility of the probe molecule and/or its crescent shape. The balance of hydrophilic/hydrophobic properties is also important, ensuring a sufficient contribution of hydrophobic interactions to binding and, at the same time, the correct strength of the concomitant aggregation of the probe. Additionally, when interacting with globular proteins, a probe should have a size suitable for binding sites of proteins (hydrophobic pockets).

Such a variety of requirements for ideal probes makes their development complicated and often relies on empirical selection from many candidates. Cyanine dyes (in particular, trimethine cyanines) can serve as such candidates.

Trimethine cyanine dyes possess attractive properties valuable for probes, such as high extinction coefficients combined with absorption in the visible spectral region (usually in the range of 500–650nm, which makes it possible to use He-Ne lasers or the second harmonic of a Nd-YAG laser for their photoexcitation), and also moderate fluorescence, depending on the medium, and the ability to noncovalently bind to various biomolecules (including characteristic H- and J-aggregates on biomolecules). The visible spectral range of registration provides an opportunity to occasionally observe color or emission changes upon the interaction of trimethine cyanines with biomolecules in solutions with the naked eye (e.g., upon the interaction of dye **53** (DEC) with HSA). In addition, the dyes have sufficient dark and light photostability to be used as probes under these conditions [3,113,136]. All this creates a good background for the widespread use of trimethine cyanines in biophysical and biomedical research and explains the large number of works in this area. Although the proposed fluorescent dye probes are often not free from shortcomings (for example, trimethine cyanine probes for DNA, proposed in [27], are prone to strong aggregation in aqueous and aqueous buffer solutions), progress in this direction is undeniable. Further development of new, convenient, and efficient probes requires, in particular, a more in-depth study of the fundamental structure–property relationship.

Currently, a large number of works are being published on the study of the photophysical (spectral-fluorescent) properties of cyanines and their changes upon noncovalent interaction with biomolecules, which is important for the development of spectral-fluorescent probes for biomolecules. Nevertheless, the need is noticed for the development of RNA-specific dyes, particularly probes for the short RNAs involved in regulating gene expression [137]. There is also a deficiency of protein-specific dye probes for more particular tasks such as the detection of proteins of certain families or folding types [138].

At the same time, much less work is devoted to the study of elementary photochemical processes in molecules of cyanines (in particular, trimethine cyanines) complexed with biomolecules: *trans*–*cis* (or *cis*–*trans*) photoisomerization and reverse *cis*–*trans* (or *trans*–*cis*) thermal isomerization of a photoisomer, formation and decay of the triplet state, and quenching of the triplet state by quenchers of various nature. Such studies can obtain valuable information on the localization of a dye probe in the biomacromolecular matrix and on the structural and energetic features of the probe–biopolymer complex. For example, the increased photoisomerization of some trimethine cyanine dyes complexed with DNA shed light on the shape of the *S*_1_ potential curve of the dyes [110]; quenching of the triplet state of the complexed dye by different quenchers provided information on the type of the complex and shielding of the complexed dye molecule from the quenchers [112]. An important property of trimethine cyanine dyes is the increase in the yield to the triplet state upon binding to biomolecules. The triplet dye molecule, when quenched by molecular oxygen, can produce singlet oxygen, which has a damaging effect on biomolecules and cells (photodynamic therapy). Since the quantum yield to the triplet state for trimethine cyanines in solution is usually close to zero, this makes it possible in principle to use trimethine cyanines for selective photodynamic therapy, when only the bound dye molecule (for example, that bound to DNA in a tumor) has a damaging effect, whereas the unbound molecule is inactive.

Thus, along with the study of the spectral and fluorescent properties of trimethine cyanine dyes complexed with biomolecules, the study of elementary photochemical processes in dye–biomolecule complexes, which is still underdeveloped, is very important.

## Data Availability

Data sharing is not applicable to this article.

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
