# Peer review of "Photonics of Trimethine Cyanine Dyes as Probes for Biomolecules"

_molecules, 2022, doi:10.3390/molecules27196367_

Round 1

Reviewer 1 Report

Сyanine dyes are widely used as fluorescent probes in biophysics and medical biochemistry. The works devoted to the study of the noncovalent interaction of trimethine cyanine dyes with biomolecules and changing the properties of these dyes upon the interaction are reviewed. This is very valuable, however, some important issues should be illustrated more clearly. Therefore this paper could be accepted after a major revision.

Questions and revision suggestions:

(1) How does dye interact with DNA and nucleic acid? How does the spectrum change? The author should add a legend for better illustration.

(2) The context of the review is relatively confusing, especially the connection and relationship between examples, and the ideas throughout the whole manuscript are suggested to be systematically combed.

(3) Compound names represented by numbers should be bold.

(4) There should be more detailed discussion about practical application of trimethine cyanine dyes in biophysics and biochemistry.

(5) There should be space between the number and the unit.

Author Response

Questions/ and revision suggestions and Authors' response:

(1) How does dye interact with DNA and nucleic acid? How does the spectrum change? The author should add a legend for better illustration.

Authors' response

The unique structure of the DNA molecule, as well as the structural features of cyanine dyes, determine the possibility of various types of dye–DNA interactions and different types of complexes. Upon the interaction of monomeric cyanine dyes with the DNA, two main types of binding are possible: the formation of intercalation complexes and binding in the groove of the DNA double helix. Intercalation requires some change in DNA configuration (adjacent base pairs create a binding pocket for the ligand). Minor groove binders, on the other hand, usually should have some flexibility since this allows the molecule to adjust its structure to follow the groove as it twists around the central axis of the helix. As well, we introduced figures illustrating dye–DNA binding modes: groove binding and intercalation (Figure 2), obtained by molecular docking.

The most characteristic effect in the absorption spectra upon dye – nucleic acids interaction is a bathochromic shift. We added figures (Figures 3a, 3b) illustrating the spectral changes and fluorescence growth upon the interaction of a representative trimethine cyanine dye (Cyan 2) with DNA, and a paragraph explaining the long-wavelength shift in the absorption spectra upon the interaction with DNA:

“The most characteristic effect observed upon the interaction of a dye molecule with nucleic acids is a bathochromic shift in the absorption spectra. The reason for this shift is changes in the solvate shell of dye molecules during complex formation, in which new components are included in the hydrate shell of the dye molecule, namely, parts of some structural units of DNA, which can be comparable to the effect of solvent replacement. Hence, one of the stages of the DNA binding process is the transfer of dye molecules from the polar phase of the solvent (water) to the less polar phase of DNA molecules, which is energetically favorable. Changes in the absorption and fluorescence spectra of a trimethine cyanine dye in the presence of DNA and the fluorescence growth are shown in Figures 3a, 3b.”

Trimethine cyanine dyes can also assemble into ordered supramolecular aggregates (in particular, H- and J-aggregates) on a DNA macromolecule as a template (see ref. [23]), which can be used for structural studies of nucleic acid molecules. The formation of aggregates can be traced in the absorption spectra (the decrease in the main band and the growth of the aggregation bands) and in the fluorescence spectra. In many cases, cyanine dyes are also characterized by the concomitant formation of disordered aggregates (with diffuse absorption spectra), which usually hinders the use of the dyes as biomolecular probes.

(2) The context of the review is relatively confusing, especially the connection and relationship between examples, and the ideas throughout the whole manuscript are suggested to be systematically combed.

Authors' response

We inserted a number of bridging sentences between the descriptions of different dyes, which we think make the review more consolidated. In these sentences, in particular, the importance of studying cis-trans equilibrium for trimethine cyanines is emphasized, as well as the important unique property of cyanines to form J- and H-aggregates is noticed.

(3) Compound names represented by numbers should be bold.

Authors' response

We checked the text of the manuscript. Now compound names represented by numbers are in bold.

(4) There should be more detailed discussion about practical application of trimethine cyanine dyes in biophysics and biochemistry.

Authors' response

We have tried to expand the section of the review on practical applications (6. Practical Application of Trimethine Cyanine Dyes in Biophysics and Biochemistry) as much as possible within the framework of our rather short review. The section devoted to the practical application of trimethine cyanine dyes in biophysics and biochemistry has been supplemented with some new examples. In brief: Cyanine dyes are used in various analytical fields, e.g., in microscopy (superresolution fluorescence microscopy), for fluorescent staining of DNA/proteins, etc. The reviews cited in this section (in particular, [113], [135]) are also devoted to the practical application of cyanine dyes.

(5) There should be space between the number and the unit.

Authors' response

We checked the text of the manuscript and corrected typos.

Reviewer 2 Report

In view of the unique photophysical and photochemical properties of trimethine cyanine, this manuscript summarizes the influence of DNA, other nucleic acids, proteins and other biological small molecules on the spectrum of trimethine cyanine. At the same time, the effects of biological small molecules on the photoisomerization and triplet states of trimethine cyanine are explained. The manuscript is well organized and instructive for trimethine cyanine-based fluorescent dyes and probe. I suggest that it can be published in this journal after minor revision.

1.     The spectral property, detection limit, quantum yield, practical application of all the probes are not mentioned. Maybe the author should present all these data with a concise table.

2.     The title is “Photonics of Trimethine Cyanine Dyes as Probes for Biomolecules”, so some biological applications of these probes should be discussed.

Author Response

Questions/ and revision suggestions and Authors' response:

(1) The spectral property, detection limit, quantum yield, practical application of all the probes are not mentioned. Maybe the author should present all these data with a concise table.

Authors' response

Spectral properties, limits of detection (LOD, LOQ), quantum yields of fluorescence in the free state and in the state bound to a biomolecule are the most important characteristics of probe dyes. Unfortunately, not all the works cited by us present these characteristics. Some authors, in particular our Ukrainian colleagues, in their works often limit themselves to indicating the relative growth of fluorescence (without determining the quantum yield of fluorescence). The spectral properties of dye probes are often not given numerically (only as spectra in the figures). Experimental data may not be available or available upon request. This significantly complicates the collection and systematization of the characteristics of probe dyes given in the literature. The authors have compiled the corresponding tables, providing data on the maxima of the absorption spectra, fluorescence, binding constants and detection limits (given in the text and in the supplementary).

(2) The title is “Photonics of Trimethine Cyanine Dyes as Probes for Biomolecules”, so some biological applications of these probes should be discussed.

Authors' response

The use of spectral-fluorescent probes based on trimethine cyanine dyes in biology, biophysics, and biochemistry is discussed in the corresponding section (6. Practical Application of Trimethine Cyanine Dyes in Biophysics and Biochemistry). In brief: Cyanine dyes are used in various analytical fields, for example, as contrast agents in microscopy, in particular, superresolution fluorescence microscopy based on single-molecule tracking. Trimethine cyanines are used for fluorescent staining of DNA, for protein staining in gel-based proteomic analysis. Some trimethine cyanine dyes are also used as PDT-agents. We have tried to expand and supplement this section with some new examples. For more information: the reviews cited in this section (in particular, [113], [135]) are devoted to the practical application of cyanine dyes. A detailed description of all aspects of the use of cyanine dyes in biology and medicine is beyond the scope of our review.

Round 2

Reviewer 1 Report

The revision is OK. The manuscript can be accepted.